# Modification of the Surface Composition of PTB7-Th: ITIC Blend Using an Additive

**DOI:** 10.3390/molecules27196358

**Published:** 2022-09-26

**Authors:** Amira R. Alghamdi, Bradley P. Kirk, Guler Kocak, Mats R. Andersson, Gunther G. Andersson

**Affiliations:** 1Flinders Institute for Nanoscale Science and Technology, Flinders University, P.O. Box 2100, Adelaide, SA 5001, Australia; 2Flinders Microscopy and Microanalysis, College of Science and Engineering, Flinders University, P.O. Box 2100, Adelaide, SA 5042, Australia; 3Department of Physics, College of Science, Imam Abdulrahman Bin Faisal University, P.O. Box 1982, Dammam 31441, Saudi Arabia

**Keywords:** *p*-anisaldehyde, interfaces in photovoltaic devices, bulk heterojunction, donor and acceptor

## Abstract

We investigated the effect of adding *p*-anisaldehyde (AA) solvent to the ink containing poly[[2,60-4,8-di(5-ethylhexylthienyl)benzo[1,2-b:3,3-b]dithiophene][3-fluoro-2[(2-ethylhexyl)carbonyl]thieno[3,4-b]thiophenediyl]](PTB7-Th) and 3,9-bis(2-methylene-(3-(1,1-dicyanomethylene)-indanone))-5,5,11,11-tetrakis(4-hexylphenyl)-dithieno[2,3-d:20,30-d0]-s-indaceno[1,2-b:5,6-b0]-dithiophene(ITIC) on the morphology of the active layer. The present study focuses on determining the effect of the additive on the compositions at the surface of the PTB7-Th: ITIC composite and its morphology, forming one side of the interface of the blend with the MoO_X_ electrode, and the influence of the structural change on the performance of devices. Studies of device performance show that the addition of the additive AA leads to an improvement in device performance. Upon the addition of AA, the concentration of PTB7-Th at the surface of the bulk heterojunction (BHJ) increases, causing an increase in surface roughness of the surface of the BHJ. This finding contributes to an understanding of the interaction between the donor material and high work function electrode/interface material. The implications for the interface are discussed.

## 1. Introduction

Organic photovoltaic (OPV) devices consist of organic-based electron donor and acceptor materials that have well-designed molecular structures, appropriate energy levels, and wide absorption in the visible spectrum, which has demonstrated a power conversion efficiency of greater than 19% [1,2]. However, by optimizing the morphology of the bulk-heterojunction structure (BHJ), a high-performance OPV can be achieved more readily. With the optimization of the BHJ structure, a sufficient interfacial contact between the donor and acceptor components of the BHJ is achieved, allowing for adequate exciton dissociation into free charges [3]. Control of the morphology, which influences the domain size and component distribution across the interface, can then be achieved [4].

Many factors can influence said morphology, including the selection of solvents and additives used in the solution preparation, donor/acceptor (D/A) ratio, and the thermal and solvent conditions for annealing the films [5]. It has been demonstrated that a significant contribution to an effective charge transport comes from the interconnectivity of the donor and acceptor phases and a proper phase separation between them which can lead to improved charge separation [6,7,8]. Furthermore, tuning the crystallinity of the active layer [9] and the domain size could affect the charge carrier mobility. This can be achieved by introducing thiol groups [10], incorporating electron-withdrawing groups in the thiophene polymers [11], or through the use of different solvent mixtures [12,13].

To improve the efficiency of the device performance through the manipulation of the BHJ, many studies have focused on additives in the solvents which affect the aggregation of active materials, and this ultimately influences the film morphology and device efficiencies [14]. Gurney et al. studied the effect of solvent vapor annealing on the performance of devices based on poly[[2,60-4,8-di(5-ethylhexylthienyl)benzo[1,2-b:3,3-b]dithiophene][3-fluoro-2[(2-ethylhexyl)carbonyl]thieno[3,4-b]thiophenediyl]](PTB7-Th):3,9-bis(2-methylene-(3-(1,1-dicyanomethylene)-indanone))-5,5,11,11-tetrakis(4-hexylphenyl)-dithieno[2,3-d:20,30-d0]-s-indaceno[1,2-b:5,6-b0]-dithiophene(ITIC). They found that the annealing process of the solvent impacts on the morphology of the BHJ, which resulted in a better phase separation of the BHJ [15]. A similar study was conducted by Wang, who investigated the influence of the solvents on the crystallinity of the acceptor (ITIC) phase. Wang achieved an improvement in the open circuit (V_oc_) and the film morphology (PTB7-Th: ITIC) [16].

Typically, solvent additives have a higher boiling point than the main solvent. They are able to dissolve at least one of the two blend components [17]. A number of additive studies have been conducted with alkane dithiols and halogenated alkanes [10]. In most polymer: fullerene systems, halogenated solvent additives such as 1,8-diiodooctane (DIO) are added to the active layer ink prior to deposition. The addition of DIO has been found to dissolve 6,6-phenyl-C-butyric acid methyl ester (PCBM) selectively and to achieve better phase separation and domain purity, thus improving device performance [18,19,20,21]. The addition of DIO to a system that uses non-fullerene acceptors, such as PTB7-Th: ITIC, results in large domains in the BHJ, which increases the probability of charge-recombination, and thus a reduction in performance [22,23]. This indicates that the processing methods must be tailored to the specific properties of the material. In addition to affecting solution miscibility, the high boiling point of DIO limits solvent evaporation and drying time, leading to a more ordered and crystalline film [4,19].

A direct comparison of PTB7-Th devices with either PC_71_BM or ITIC acceptors resulted in an outcome that indicated that 3 wt.% DIO optimized the domain sizes and purity in the fullerene system, whereas similar conditions yielded stronger separated phases in the non-fullerene system [23,24]. In another study, Zhao et al. discovered the benefits of using DIO as a solvent additive in other non-fullerene systems. Their findings showed that adding 0.5 wt.% DIO contributed to the creation of a favorable aggregation, leading to an improvement in charge transfer, low recombination, and high performance, which enabled poly[(2,6-(4,8-bis-5-(2-ethylhexyl)thiophen-2-yl)benzo[1,2-b:4,5-b′]dithiophene)-co-(1,3-di(5-thiophene-2-yl)-5,7-bis(2-ethylhexyl)benzo[1,2-c:4,5-c′]dithiophene-4,8-dione)](PBDB-T):ITIC to outperform its fullerene counterpart [25]. Hence, it is possible that the orientation of the alkyl side chains found on the IDT backbone interact with the DIO, which results in better miscibility by lowering steric hindrance between the molecules, thereby enabling finer phase separation and creating purer crystalline phases [26]. However, according to Zhan et al., different mixing enthalpies render the control of donor polymers mixing with non-fullerene acceptors (NFAs) difficult [27]. This limitation hinders the realization of an optimal microstructure for enabling pure and mixed domains in NFA systems with solvent additives [28,29,30]. Therefore, typical optimization approaches adopted in fullerene-based procedures may lack validity for NFA solar cells.

In spite of the advantages of using high boiling point solvent additives, the slow process of evaporation means slower fabrication times, due for example to the need to hold the device under vacuum for long periods of time to eliminate residual solvent. Any remaining residual solvent within the active layer could result in the formation of radicals under UV illumination [31] or the creation of a pathway for oxygen penetration, thereby resulting in film degradation [32,33].

A further factor influencing the device performance is the phase segregation and concentration distribution of the donor and acceptor [34,35,36,37,38]. In simple terms, the transmission of a hole along consecutive donor domains will benefit from interfaces that are donor-rich at the anode and vice versa. As an example, it was proposed that increasing the surface coverage of poly(3-hydroxythiophene) (P3HT) at the anode would result in an improved conventional OPV; however, device performances were not strictly based on the measured vertical stratification [13,39].

In the literature, BHJs are reported to have a high donor concentration at the surface in the inverted OPV. Chen et al. investigated the effects of the fluorinated counterparts on the concentration gradient, and they explored the impact of a higher concentration of donor in the BHJ before and after fluorination. The fluorination of NFAs, however, reduces the number of donors in this donor-rich region. As a result, encouraging the mixing of donors and acceptors leads to the generation of efficient charges [40]. Although the concentration distribution has been well studied, the identification of the composition of blends at the surfaces has not been studied with the same rigor. We anticipate that the donor which facilitates the hole transfer to the high work function electrode should be enriched at the interface with the high work function electrode, and this would have a significant impact for our understanding of the functions of BHJ-based OPVs. Huang et al. investigated the effect of using *p*-anisaldehyde (AA) on the device performance, based on poly[(2,6-(4,8-bis-5-(2-ethylhexyl)thiophen-2-yl)benzo[1,2-b:4,5-b′]dithiophene)-co-(1,3-di(5-thiophene-2-yl)-5,7-bis(2-ethylhexyl)benzo[1,2-c:4,5-c′]dithiophene-4,8-dione)]:poly{[N,N0-bis(2-octyldodecyl) naphthalene1,4,5,8-bis(dicarboximide)-2,6-diyl]-alt-5,5′-(2,2′-bithiophene)}(PBDB-T:N2200). They found that the interfacial contact between PBDB-T and N2200 itself and between the active layer and PEDOT: PSS improved, which led to the promotion of efficient exciton dissociation [41].

In the present work, we modified the compositions of the active layer by introducing AA as an additive. AA has both an oleophilic methoxy group and a hydrophilic aldehyde group. The purpose of adding AA was to optimize the interfacial compatibility of PTB7-Th and ITIC by increasing the PTB7-Th concentration at the interface. AA is non-toxic and has a high volatility, enabling its complete evaporation during the spin coating process of the active layer [42]. The focus of this study is to understand the composition of the surface and near-surface area of the blend based on PTB7-Th: ITIC and the effect of AA additive. In inverted devices, it is necessary to have a sufficient electron donor concentration at the high work function electrode. Since the energy levels between the donor (PTB7-Th) and MoO_X_ are rather close, the donor enrichment at the interface promotes the charge transport to MoO_X_. In our work, we applied neutral impact collision ion scattering spectroscopy (NICISS) as a depth profiling technique to determine quantitatively the concentration depth profiles at the surface with a depth resolution of a few Å, and thus to determine the concentration of materials present at the surface. We report the effect of the additive AA solvent on the composition of the sample at the surface, and it effect on the surface morphology. We show that AA has a beneficial effect on phase separation, leading to an increase in the donor concentration at the surface and to improved device performance.

## 2. Results and Discussion

### 2.1. Device Performance

Prior to the BHJ investigation, and prior to deposition on the performance of the OPV devices, it was important to determine the influence of the addition of AA to the active layer ink. Comparisons of the performance of the devices with and without the AA additive can be found in Table 1. To reduce the potential traces of AA that could still be present in the active layer, we used high vacuum drying (10^−6^ mbar) as a surface/morphology treatment during fabrication. This is an outcome of the work with the PTB7-Th: ITIC system reaching a higher efficiency by using a more environmentally friendly solvent system and a novel drying technique for the thin film. The common procedure for drying the BHJ layer and removing any residue of the additive is by annealing the film at a specific temperature [40,42].

The addition of AA to the active layer prior to the deposition of the active layer resulted in an improvement of PCE from 7.03% to 8.20%. This increase predominately resulted from the increase in J_SC_ and FF, whereas a minimal change was observed for the open circuit voltage (V_OC_).

In the literature, it has been shown that adding solvent additives to active layer inks improves the morphology and results in an improvement in the J_SC_ and FF [43,44,45]. By achieving an improved performance, there is a reduction in the probability of an electron-hole recombination, leading to an increase in J_SC_ and FF [44]. Concerning our results, shown in Table 1, these findings demonstrate that AA, which has an oleophilic methoxy group and a hydrophilic aldehyde group, is a very effective additive to the here chosen BHJ and that it achieves excellent performance in an OPV. A comparison table of the surface roughness and concentration of BHJ with/without AA will be discussed below.

### 2.2. AFM Results

To get a general understanding on the influence of the AA additive on the PTB7-Th: ITIC morphology, atomic force microscopy (AFM) was used to investigate the surface morphology of the active layer. It has been speculated that changes observed on the surface of the BHJ could be related to changes to the morphology [46]. AFM images of the surface of spin-coated PTB7-Th: ITIC with and without AA additive are shown in Figure 1.

Based on the AFM images and calculated roughness values, significant changes to the surface of the BHJ were observed when comparing active layers with and without AA prior to deposition. Firstly, the addition of AA resulted in an increase in feature size, as well as an increase in surface roughness, indicating a likely increase in domain size. This increase in roughness would usually result in an increased probability in charge recombination, as seen in previous literature that prepared PTB7-Th: ITIC devices with the addition of DIO [24]. It is possible that the increase in roughness is related to the enhancement of polymer order and the crystallinity direction of the PTB7-Th [47,48]. In short, this could lead to improved exciton dissociation and charge transport in the photoactive film.

### 2.3. NICISS Results

Figure 2 shows the NICIS-TOF spectra of layers of pristine PTB7-Th, pristine ITIC, a blend (PTB7-Th: ITIC) with 2% AA, and a blend (PTB7-Th: ITIC) without the additive. Vertical lines are an indication of the onset of the signal of the He projectiles backscattered from the elements constituting the sample. The heavier elements have higher kinetic energy, thus the backscattering of projectiles was detected at lower TOF. The spectrum of PTB7-Th contains contributions from sulphur (S), fluorine (F), oxygen (O) and carbon (C). Additionally, a small signal of silicon was obtained in the PTB7-Th spectra, implying the presence of an impurity, most likely siloxane [49,50]. The spectrum of ITIC contained S, O, N, and C. These elements are the main components of ITIC. S, O, and C can be identified in the blend spectra with and without the additive. Mainly, the S at the BHJ showed enrichment at the surface, which can be identified through the enhanced count rate at the onset of the step related to S. However, F and N could not be identified in the NICIS spectra as their count rates are too low to allow a proper evaluation.

In order to determine the composition of the BHJ at its surface, we evaluated the S concentration depth profiles quantitatively for the BHJ samples with and without the AA additive. As a first step, the TOF spectra for the BHJ were converted into concentration depth profiles as described in reference [51]. To convert count rate into concentration requires knowledge of the bulk concentration of one of the elements. In the present case, the concentration of carbon has been used in combination with the C:S intensity ratio as measured in the pristine PTB7-Th and ITIC. This allowed us to determine what the S count rate in the spectra of the BHJ should be based on the known bulk composition of the BHJ (1:1.3 ratio of PTB7-Th to ITIC). The count rate for the known bulk concentration was used as a reference and set to unity. This relative measure for the concentration is used for the *y*-axis in Figure 3.

In order to understand the distribution of the species forming the layers at the near-surface region, we determined the concentration depth profile of S in the BHJ with and without the additive, as shown in Figure 3. In the measured profile, the zero-depth refers to the outermost layer where the increase in the depth indicates the direction toward the bulk. Both depth profiles of the S have the same onset close to 0 Å, and the ratio of the profiles in the subsurface is very much the same.

Figure 3 indicates a higher concentration of S at the surface compared with the region below the surface (i.e., at depth > 40 Å). This is indicated by an increase in the concentration up to 40 Å. We attribute the enhancement of the S to a greater presence of the PTB7-Th at the surface of the BHJ than in the subsurface region (depth > 40 Å). This analysis of the concentrations is based on the chemical structure of PTB7-Th and ITIC, with the S concentration being higher in the PTB7-Th component than in the ITIC. However, the distribution of the compositions throughout the BHJ is more complicated. The concentration of PTB7-Th in the subsurface is depleted compared with the surface.

The presence of the PTB7-Th, whether with the additive or without the additive, did not reach unity, i.e., the related bulk concentration (1:1.3) shown by the dashed line in Figure 3. The values of the surface concentration can be seen in Table 2. Hence, PTB7-Th is depleted at the surface of the BHJ in both systems and thus must have segregated more to the substrate/BHJ interface. Therefore, the ITIC shows an overall enrichment in the surface region compared with the PTB7-Th. This means that the PTB7-Th does not occupy the whole outermost layer. A possible reason for the depletion of the PTB7-Th at the surface is the drying behaviors for the two components. The drying process during the production of the spin-coated films could affect the distribution and lead to poor miscibility or an incompatible crystalline structure of the PTB7-Th and ITIC, and thus also influence the formation of the crystalline structure. It is well known in the literature that the solubility of materials within the active layer ink plays an important role in the control of morphology, thus, is a key factor when selecting appropriate base solvents and additives [52,53]. In our studies, we suspected that the PTB7-Th would remain in the solution longer than ITIC, giving the appearance of the ITIC drying faster than PTB7-Th. This was based on the solubility of ITIC, which is lower than that of PTB7-Th. As a result, the ITIC moves into the solid phase sooner and precipitates on the surface first. PTB7-Th remains quenched if no additive is used to transform the blend into the equilibrium crystalline phase.

The second phenomenon is related to the difference of the ratio between the surface and the subsurface, which is based on the surface energy of the two components. The depth profile in Figure 3 shows a notable difference in the concentration of PTB7-Th at the surface in the first few Å (~0 to 40 Å) and at subsurface (~50 to 100 Å). This indicates that there is a higher concentration of PTB7-Th at the surface than in the subsurface. As reported by Lin et al., ITIC has a higher surface energy than PTB7-Th, which allows PTB7-Th to move toward the surface [54]. A similar result was reported by Wang et al., who found a strong segregation of the components PTB7-Th, PC71BM, and m-ITIC in the ternary system. The authors determined the surface energy and the wetting coefficient for the components. The PTB7-Th was the component with lowest surface energy located at the surface, and the ITIC tended to be segregated between the PTB7-Th and the PC71BM [55]. However, their findings included only an estimation of the surface enrichment. The authors also did not show a detailed profile distinguishing between surface and subsurface.

After adding 2% of AA into the BHJ, the surface shows an increase in PTB7-Th concentration at the surface. For a schematic showing this enrichment, see Figure 4. A possible reason for this phenomenon could be the change in the bulk properties of the BHJ upon adding the AA. The AA has an oleophilic methoxy and a hydrophilic aldehyde group, and it is a polar molecule with high surface energy which could drive the electron donor to the surface.

In the inverted devices, the energy levels between the donor (PTB7-Th) and MoO_X_ are closer, thus donor enrichment at the surface is beneficial for charge transport and collection for an inverted OPV. However, an enrichment of accepter in the surface region would result in less intermixing of donors and acceptors and a decrease in the interfacial area for exciton dissociation, which would result in reduced device performance.

## 3. Experimental Procedure: Material and Device Fabrication

### 3.1. Materials

PTB7-Th (Mn = 80,000) was purchased from 1-Materials Inc and the acceptor polymer ITIC was purchased from Raynergy Tek Inc. o-Xylene and *p*-anisaldehyde (AA) were purchased from Sigma-Aldrich, St. Louis, MO, USA, whereas acetone and 2-propanol were supplied by Chem-Supply. All solvents were used directly without purification.

### 3.2. Device Fabrication

The substrates were cleaned by immersing them in a 5% Pyroneg solution for 20 min at 90 °C. Subsequently, the substrates were rinsed with MilliQ water and sonicated for ten minutes each in MilliQ water, acetone, and 2-propanol. The cleaned substrates were dried under a stream of nitrogen, followed by 20 min of UV/ozone treatment.

The ZnO solution was prepared according to previously published procedures [56]. Zinc acetate dihydrate (500 mg) (Sigma-Aldrich, 99.9%) and ethanolamine (140 mg) (Sigma-Aldrich, 99.5%) were dissolved under vigorous stirring overnight in 2-methoxyethanol (5 mL) (Sigma-Aldrich, 99.8%). The solution was filtered with a PTFE syringe filter (0.45 μm) prior to being spun-coated on a cleaned ITO-coated glass substrate at 3000 rpm for 60 s, producing a thin film of ~25–30 nm. The resulting film was then annealed in a 280 °C pre-heated furnace in air for 10 min.

For the BHJ blend films of the devices, PTB7-Th: ITIC inverted polymer solar cells were fabricated using non-toxic solvent additive AA and host solvent o-xylene. Devices were fabricated in a glass/ITO/ZnO/PTB7-Th: ITIC/MoO_X_/Ag device configuration. The active layer ink was prepared by dissolving active layer materials, with a donor: acceptor weight-to-weight ratio of 1:1.3, in o-xylene (total 19.5 mg mL^−1^), either with or without 2% *v*/*v* AA. The AA additive has been tested in the literature for a different system, and it recorded the highest efficiency at 2% [42]. Here, we choose the same AA concentration. The active layer ink was mixed overnight at 75 °C under vigorous stirring. With the prepared ink, the solution was spin-coated over the ZnO film for 60 s at 2000 rpm and 2500 rpm, for the controlled and modified devices, respectively. The structures of PTB7-Th, ITIC, and AA are illustrated in Figure 5.

After the spin-coating of the active layer, the devices were vacuum dried in an evaporation chamber at 10^−6^ mbar for a further hour after the chamber pressure had reached 10^−6^ mbar. After vacuum drying, MoO_X_ and Ag were deposited via the following method. The MoO_X_ (12 nm) was thermally deposited on the BHJ layer under high vacuum using a Covap thermal evaporation system (Angstrom Engineering). This was followed by the evaporation of the Ag electrode (80 nm) using a shadow mask to define the active area to 0.1 cm^2^.

### 3.3. Characterization

#### 3.3.1. Device Performance

The solar cell devices were measured using an Oriel Solar simulator fitted with a 150 W Xeon lamp (Newport), filtered to give an output of 100 mW cm^−2^ at AM 1.5 (air mass) standard and calibrated using a silicon reference cell with NIST traceable certification. Device testing was conducted under ambient conditions. The averages were based on 6 inverted devices with a defined device area of 0.1 cm^2^.

#### 3.3.2. Neutral Impact Collision Ion Spectroscopy (NICISS)

NICISS is an analytical technique for characterizing the concentration depth profile of the elements at soft matter surfaces. NICISS yields information about the concentration depth profile of specific elements in a sample with an average thickness of up to about 30 nm at a depth resolution of a few Angstrom (Å) [57]. The sample is bombarded with a pulsed beam of helium ions (He^+^) with kinetic energies of 1–5 keV [58,59]. The He projectiles are backscattered from the target and the time of flight (TOF) to the detector is measured. In this backscattering process, the loss in kinetic energy of the projectile is a function of the depth of the atom from which the projectile is backscattered. There are two types of energy loss processes. The first type of loss process is elastic energy loss, which enables the determination of the mass of the elements. The other type of loss process is inelastic energy loss, whereby the projectiles lose energy on their trajectory through the bulk by low angle scattering and electronic excitations (stopping power) [60]. This type of loss can be used to determine the concentration depth profiles of the elements in the sample because it governs the depth travelled by the projectiles, provided the target atom is heavier than the projectile. A concentration depth profile of hydrogen thus cannot be determined. Instead, a recoil hydrogen background is found in the spectra and removed in the data analysis. More details about the method for determining the concentration depth profiles can be found in reference [61]. The NICISS results are presented as a spectrum consisting of individual peaks and steps representative of the different elements in the sample. The first peak in the NICISS spectra is the photon peak which corresponds to the first He^+^ interactions with the sample surface.

In non-deconvoluted NICISS, a concentration depth profile count rate can be found at negative depth. This count rate at negative depth does not have any physical meaning in the sense of concentration at negative depth. The effect is related to the finite energy resolution of the method and is explained in detail elsewhere [62,63]. The elements investigated in this work are sulphur, fluorine, oxygen, nitrogen and carbon. A previous investigation by our lab using NICISS determined the composition of the surface and near surface area of a blend of P3HT: PCBM, identifying a layered structure at the surface [49]. The known bulk concentration of a sample is used to convert the measured count rate into concentration [51].

#### 3.3.3. Atomic Force Microscope (AFM)

The BHJ layer topography is crucial for achieving high PCE solar cells. Atomic force microscopy (AFM) was run in tapping mode to study the topography. In AFM tapping mode a tip is attached to an oscillating cantilever which scans the surface of a sample. Interactions between the tip and the surface are registered as deviations of the oscillating pattern of the cantilever. These deviations are monitored with a laser as the surface is scanned, and then transferred into a three-dimensional topography map.

## 4. Conclusions

The influence of AA as an additive on the ratio of PTB7-Th: ITIC as a function of the depth was investigated by measuring the concentration depth profile of sulphur (S). The depth profiles of the S were measured directly to determine the concentrations of the compositions at the surface and in the subsurface region. The results showed an overall enrichment of ITIC at the top layer of the BHJ, followed by the PTB7-Th layer, revealing an enriched layer of PTB7-Th at the substrate/BHJ interface. The difference in the ratio of the PTB7-Th at the surface and the subsurface region is related to the difference of the surface energy of PTB7-Th and ITIC. PTB7-Th contains lower surface energy materials, driving the PTB7-Th toward the surface. The AA additive increased the concentration of PTB7-Th at the surface and decreased the concentration of ITIC. The enhancement of PTB7-Th at the surface is beneficial for the charge transfer to the MoO_X_ electrode in the inverted OPV. AFM results also showed that the addition of AA increased the feature size and the roughness of the BHJ, which improved exciton dissociation and charge transport in the photoactive film. Consequently, the enhancement of the donor at the surface after adding AA results in the enhancement of the performance of the devices. An increase in PCE from 7.03% to 8.20% was achieved after adding AA to the BHJ film.

## Figures and Tables

**Figure 1 molecules-27-06358-f001:**
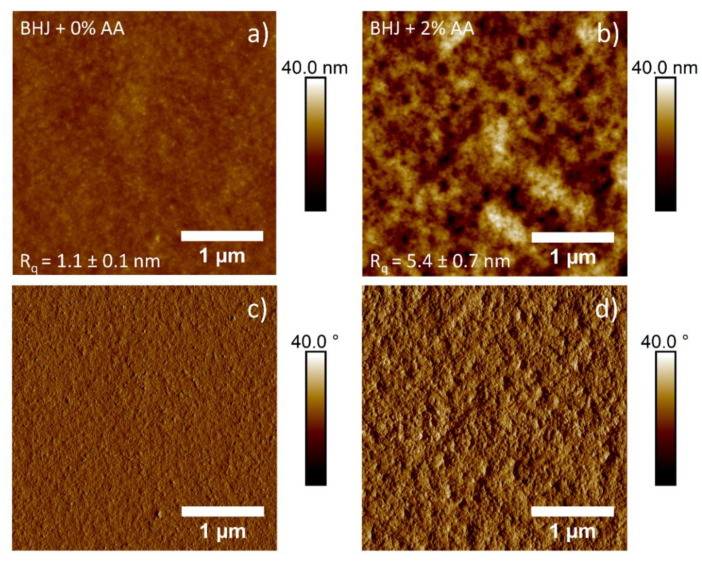
AFM topography height (**a**,**b**) and phase (**c**,**d**) images (5 µm × 5 µm) of the surface morphology of the BHJ with 0% AA (**a**,**c**) and 2% AA (**b**,**d**). Average R_q_ roughness was calculated from 5 scan locations per sample.

**Figure 2 molecules-27-06358-f002:**
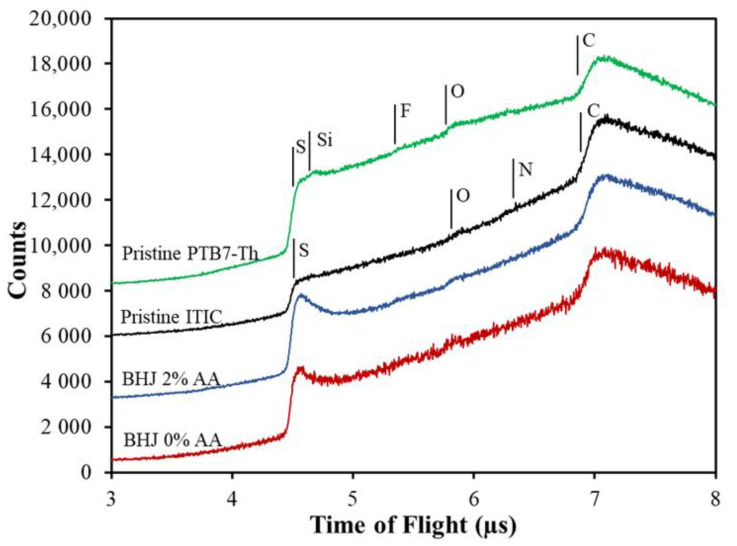
NICIS TOF spectra of PTB7-Th, ITIC, and a 1:1.3 blend of PTB7-Th and ITIC. Signal onset of helium backscattered from sulphur, silicon, oxygen, nitrogen, and carbon is marked by vertical bars. The spectra are offset vertically for clarity.

**Figure 3 molecules-27-06358-f003:**
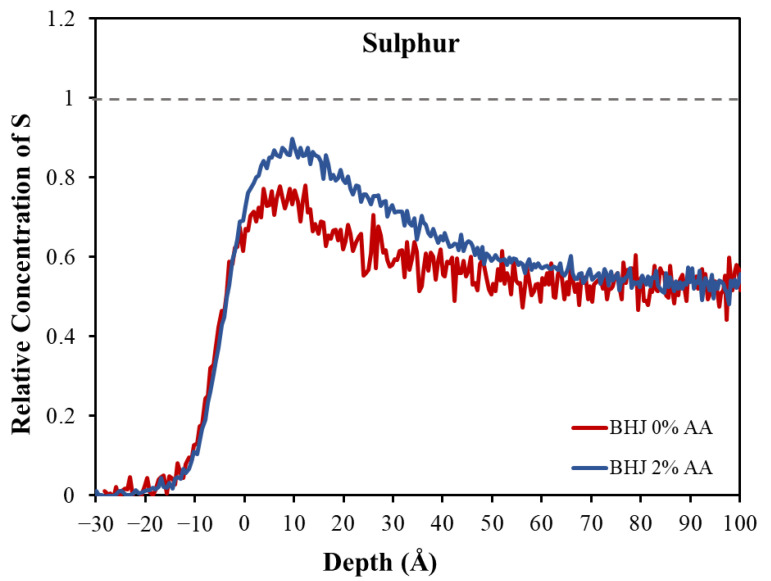
Comparing the concentration profiles of BHJ with 2% AA and without the additive. The dashed line indicates the bulk concentration of S for the ratio of 1:1.3 (the ratio of PTB7-Th: ITIC). The S to C ratio for the bulk in the spectrum of the BHJ can be determined from the elemental composition of the individual components.

**Figure 4 molecules-27-06358-f004:**
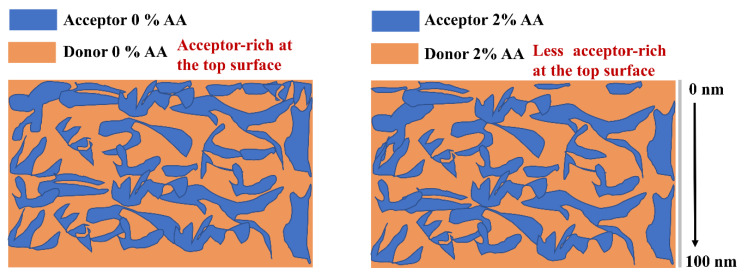
The schematic shows the effect of the AA additive on the donor enrichment at the surface.

**Figure 5 molecules-27-06358-f005:**
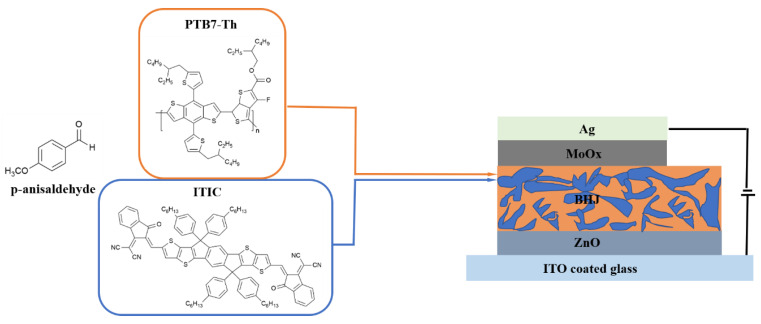
The device structure and the molecular structures of PTB7-Th, ITIC and p-anisaldehyde (AA).

**Table 1 molecules-27-06358-t001:** Device Characteristics of PTB7-Th: ITIC with and without the AA additive. The averages were based on 6 inverted devices with a defined device area of 0.1 cm^2^.

Device	J_SC_ (mA cm^−2^)	V_OC_ (V)	FF	PCE (%)
PTB7-Th: ITIC + 0% AA	15.55 ± 0.16	0.81 ± 0.01	0.56 ± 0.01	7.03 ± 0.13
PTB7-Th: ITIC + 2% AA	16.47 ± 0.18	0.80 ± 0.01	0.62 ± 0.01	8.20 ± 0.21

**Table 2 molecules-27-06358-t002:** Surface roughness and S relative concentration of the PTB7-Th: ITIC device with and without AA. A relative concentration of 1 would mean that the surface concentration is the same as the bulk concentration.

Device	Surface Roughness (nm) (AFM)	Relative S Concentration (NICISS)
PTB7-Th: ITIC + 0% AA	1.1 ± 0.1	0.78 ± 0.1
PTB7-Th: ITIC + 2% AA	5.4 ± 0.1	0.90 ± 0.1

## Data Availability

Data are available upon reasonable request.

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
