# Peer review of "Modification of the Surface Composition of PTB7-Th: ITIC Blend Using an Additive"

_molecules, 2022, doi:10.3390/molecules27196358_

Round 1
Reviewer 1 Report
The manuscript Modification of the Composition at the Surface of PTB7-Th: ITIC Blend Using an Additive to Improve the Performance of Organic Photovoltaics is written good. Some mandatory revisions are needed.
1- The abstract should be concise and brief.
2. The title should be revised as: its too long
3- Short keywords can be replaced
4- Author should use Justify intonation not the centre throughout the manuscript.
5. The introduction part can be elaborated a little more with applied industrial application, [Ref] https://doi.org/10.1016/j.molliq.2022.119765
6. Add a brief literature review of PTB7-Th: ITIC Blend in the Introduction part for various applications.
7. Language revisions is recommended throughout the manuscript. Use short sentences rather longer ones. Abstract should be in Present tense while Conclusion in the past. A proper scientific narration should be followed.
9. Figures should follow the same format, high pixel resolutions should be added. Figure 2, scall can be more closer to the figure.
11. A comparative table should be added.
12. Graphical abstract should be added.
Author Response
The manuscript Modification of the Composition at the Surface of PTB7-Th: ITIC Blend Using an Additive to Improve the Performance of Organic Photovoltaics is written good. Some mandatory revisions are needed.
1- The abstract should be concise and brief.
Reply: We followed the recommendation of the reviewer and revised the abstract accordingly.
- The title should be revised as: its too long
Reply: We have revised the title.
3- Short keywords can be replaced
Reply: We have added keywords.
4- Author should use Justify intonation not the centre throughout the manuscript.
Reply: We have revised the manuscript accordingly.
- The introduction part can be elaborated a little more with applied industrial application, [Ref] https://doi.org/10.1016/j.molliq.2022.119765
Reply: We appreciate the recommendation of the reviewer. However, the subject of the referenced article is in our opinion not related to our manuscript. (“Polyhedral Co3O4@ZnO nanostructures as proficient photocatalysts for vitiation of organic dyes from waste water”).
- Add a brief literature review of PTB7-Th: ITIC Blend in the Introduction part for various applications.
Reply: We have expanded the review of literature of PTB7-Th: ITIC in the last paragraph on page 3.
- Language revisions is recommended throughout the manuscript. Use short sentences rather longer ones. Abstract should be in Present tense while Conclusion in the past. A proper scientific narration should be followed.
Reply: We have checked and revised language and grammar. The changes are marked but not listed individually.
- Figures should follow the same format; high pixel resolutions should be added. Figure 2, scall can be more closer to the figure.
Reply: We revised all Figures in the manuscript accordingly.
- A comparative table should be added.
Reply: We have added a comparative Table at the end of page 13. Also, we have added a sentence in the first paragraph on page 14.
- Graphical abstract should be added.
Reply: We have updated the existing graphical abstract.
Reviewer 2 Report
In general, the manuscript is very well written. So, I think it is publishable in Molecules, but the figure's quality needs to improve. Figure 2 should be centralized; Figures 3 and 4 need to be standardized; it´s are of different sizes.
Author Response
In general, the manuscript is very well written. So, I think it is publishable in Molecules, but the figure's quality needs to improve. Figure 2 should be centralized; Figures 3 and 4 need to be standardized; it´s are of different sizes.
Reply: We have checked all Figures in the manuscript and revised where needed.
Reviewer 3 Report
The article titled "Modification of the Composition at the Surface of PTB7-Th: ITIC Blend Using an Additive to Improve the Performance of Organic Photovoltaics" explains how the addition of 2% of p-anisaldehyde as an additive to the blend of active materials modifies the final composition of the surface in direct contact with MoOx layer. The authors use the results of AFM and NICISS to explain the PV results. The approximation is not incorrect but the manuscript suffers from shortcomings:
1. The manuscript needs a deep review of English, it is very difficult to understand.
2. There are already studies on p-anisaldehyde that the authors do not name (ex. https://pubs.acs.org/doi/full/10.1021/acsaem.9b01624), and they are also more detailed than the work presented here.
3. Why have the authors chosen to add 2% AA? If other % have been tested, they should be included and, if not, that 2% should be justified.
4. I think it would be interesting to include J-V and EQE charts and, perhaps, an impedance study could better clarify the results, although it is not a sine qua non condition.
Author Response
The article titled "Modification of the Composition at the Surface of PTB7-Th: ITIC Blend Using an Additive to Improve the Performance of Organic Photovoltaics" explains how the addition of 2% of p-anisaldehyde as an additive to the blend of active materials modifies the final composition of the surface in direct contact with MoOx layer. The authors use the results of AFM and NICISS to explain the PV results. The approximation is not incorrect but the manuscript suffers from shortcomings:
- The manuscript needs a deep review of English, it is very difficult to understand.
Reply: We have checked and revised language and grammar.
- There are already studies on p-anisaldehyde that the authors do not name (ex. https://pubs.acs.org/doi/full/10.1021/acsaem.9b01624), and they are also more detailed than the work presented here.
Reply: We are aware of this paper, and we described and reference this paper in the introduction on page 5 in the second half of the third paragraph. However, the authors of this paper used a different donor and acceptor and a different device structure in their work. Further, they studied the effect of AA on the interface contact between the D/A and HTL based on optical properties. They showed an average estimation of the interfacial contact but did not show the exact distribution of the D/A at the interface. In the present case, we show a detailed depth profile distinguishing between the surface and subsurface of the donor and acceptor and which effect AA has on these.
- Why have the authors chosen to add 2% AA? If other % have been tested, they should be included and, if not, that 2% should be justified.
Reply: We have not tested different percentages of AA in our work. We only tested two case, 2% AA and no added AA. The reason is that the concentration effect has already been investigated for another system for which 2% AA was found to result in the most efficient device. This is described in the manuscript on page 6 in the last paragraph. While testing various AA concentrations is an interesting suggestion, we consider this as interesting future work beyond the current work. For the current work it was only important to choose an AA concentration which results in an increase in efficiency compared to the system without AA. We have added this information in the first paragraph on page 7.
- I think it would be interesting to include J-V and EQE charts and, perhaps, an impedance study could better clarify the results, although it is not a sine qua non condition.
Reply: This is an interesting suggestion and measuring EQE will be added to our future work.
Reviewer 4 Report
In this manuscript, the effect of the additive on the compositions at the surface of the PTB7-Th: ITIC composite and its morphology forming one side of the interface of the PTB7-Th: ITIC were carried out with the MoOx electrode and the influence of the structural change on the performance of devices is investigated. The corresponding experimental results showed that the AA additive can led to increase the short circuit (JSC) and fill-factor (FF), resulting in an improvement in device performance. With the addition of AA, the highest power conversion efficiency (PCE) achieved was 8.2%. The work is interesting, however, the more detailed discussions should be enhanced for better understanding the underlying reason on the effect of solvent additive. I am pleased to recommend this manuscript acceptable for potential publication in Molecules if the authors can well address the following issues.
1. The solvent additive has been confirmed as an efficient method to improve the performance of organic solar cells, however, the doping ratio or doping time before spin coating are the critical parameters for achieving efficient cells. The related reports should be included in the introduction section. Some cases were reported with two or more solvent additives in BHJ or layer-by-layer organic solar cells, such as 10.1039/d2tc00024e; 10.1016/j.cej.2022.136368; 10.1039/d2ta02914f.
2. I can not agree with the statement “The AA additive has been tested in the literature and recorded the highest efficiency with 2%.”, in fact, the PCE of cells should strongly determined by the donor and acceptor materials, rather than solvent additive.
3. The authors claim that high vacuum drying (10-6 mbar) as a surface/morphology treatment during fabrication. The more detailed information should be provided, such as how long time or the temperature.
4. In this work, the AA doping ratio is 0% or 2%, how about other doping ratios on the performance of OSCs? The series and shunt resistance of OSCs should be calculated according to the J-V curves, the more discussion on the effects on FF or PCE should be added, such as 10.1002/smll.202104215.
5. For the Figure 2, the phase image of AFM should be provided, rather than the height image. The surface morphology can not reflect the molecular arrangement or phase separation in the active layer with or without AA additive.
6. How about the AA of effect on the normal structure devices?
7. The authors should carefully revise the figures in the main text. Some figures are too large.
Author Response
In this manuscript, the effect of the additive on the compositions at the surface of the PTB7-Th: ITIC composite and its morphology forming one side of the interface of the PTB7-Th: ITIC were carried out with the MoOx electrode and the influence of the structural change on the performance of devices is investigated. The corresponding experimental results showed that the AA additive can led to increase the short circuit (JSC) and fill-factor (FF), resulting in an improvement in device performance. With the addition of AA, the highest power conversion efficiency (PCE) achieved was 8.2%. The work is interesting, however, the more detailed discussions should be enhanced for better understanding the underlying reason on the effect of solvent additive. I am pleased to recommend this manuscript acceptable for potential publication in Molecules if the authors can well address the following issues.
- The solvent additive has been confirmed as an efficient method to improve the performance of organic solar cells, however, the doping ratio or doping time before spin coating are the critical parameters for achieving efficient cells. The related reports should be included in the introduction section. Some cases were reported with two or more solvent additives in BHJ or layer-by-layer organic solar cells, such as 10.1039/d2tc00024e; 10.1016/j.cej.2022.136368; 10.1039/d2ta02914f.
Reply: We appreciate this comment and have cited the reference “10.1039/d2ta02914f” at the end of paragraph two on page 3.
- I can not agree with the statement “The AA additive has been tested in the literature and recorded the highest efficiency with 2%.”, in fact, the PCE of cells should strongly determined by the donor and acceptor materials, rather than solvent additive.
Reply: The reviewer is correct and the sentence is revised such that we now refer to a specific system for which 2% had resulted in devices with the highest efficiency.
- The authors claim that high vacuum drying (10-6 mbar) as a surface/morphology treatment during fabrication. The more detailed information should be provided, such as how long time or the temperature.
Reply: We agree with the reviewer. The devices were kept in high vacuum at 10-6 mbar for a further hour after 10-6 mbar was reached. We have revised the sentence to be clearer in the manuscript on page 7 in the second paragraph.
- In this work, the AA doping ratio is 0% or 2%, how about other doping ratios on the performance of OSCs? The series and shunt resistance of OSCs should be calculated according to the J-V curves, the more discussion on the effects on FF or PCE should be added, such as 10.1002/smll.202104215.
Reply: We have not tested different percentages of AA in our work. We only tested two case, 2% AA and no added AA. The reason is that the concentration effect has already been investigated for another system for which 2% AA was found to result in the most efficient device. This is described in the manuscript on page 6 in the last paragraph. While testing various AA concentrations is an interesting suggestion, we consider this as interesting future work beyond the current work. For the current work it was only important to choose an AA concentration which results in an increase in efficiency compared to the system without AA..
- For the Figure 2, the phase image of AFM should be provided, rather than the height image. The surface morphology can not reflect the molecular arrangement or phase separation in the active layer with or without AA additive.
Reply: We partially agree with the comment. AFM phases images are found in literature when discussing molecular arrangement or phase separation in the active layer, however, they are often shown together with AFM height images. We thus have included the AFM phases images and revised the caption of Figure 2 on page 10.
- How about the AA of effect on the normal structure devices?
Reply: There might be an effect of AA on the normal structure, however, such an investigation would be beyond the current work.
- The authors should carefully revise the figures in the main text. Some figures are too large.
Reply: We have revised the Figures in the manuscript.
Further changes:
- i) we have expanded the caption of figure 2.
Round 2
Reviewer 3 Report
This time the language has improved enough to be able to read it without problems. Be careful, since the manuscript is plenty of "Error! Reference source not found.." sentences. You need to double-check these things before sending an article. Regarding the content, I think it is suitable for Molecules.
Author Response
Reviewer comment: This time the language has improved enough to be able to read it without problems. Be careful, since the manuscript is plenty of "Error! Reference source not found.." sentences. You need to double-check these things before sending an article. Regarding the content, I think it is suitable for Molecules.
Response: these errors were introduced when our Word file was converted by the journal into a different format. We have replaced the figure and table referencing.